# Prevalence, specific and non-specific determinants of complementary medicine use in Switzerland: Data from the 2017 Swiss Health Survey

**Delphine Meier-Girard** [1] *, **Emmanuelle Lüthi**[2], **Pierre-Yves Rodondi**[2], **Ursula Wolf**[1]

**1** Institute of Complementary and Integrative Medicine, University of Bern, Bern, Switzerland, **2** Institute of Family Medicine, Faculty of Science and Medicine, University of Fribourg, Fribourg, Switzerland

* delphine.meier@unibe.ch

## Abstract

### Objectives

To determine the prevalence of use of complementary medicine (CM) in Switzerland in 2017, its development since the 2012 Swiss Health Survey, and to examine specific and non-specific sociodemographic, lifestyle and health-related determinants of CM use as compared to determinants of conventional health care use.

### Materials and methods

We used data of 18,832 participants from the cross-sectional Swiss Health Survey conducted by the Swiss Federal Statistical Office in 2017 and compared these data with those from 2012. We defined four CM categories: (1) traditional Chinese medicine, including acupuncture; (2) homeopathy; (3) herbal medicine; (4) other CM therapies (shiatsu, reflexology, osteopathy, Ayurveda, naturopathy, kinesiology, Feldenkrais, autogenic training, neural therapy, bioresonance therapy, anthroposophic medicine). Independent determinants of CM use and of conventional health care use were assessed using multivariate weighted logistic regression models.

### Results

Prevalence of CM use significantly increased between 2012 and 2017 from 24.7% (95% CI: 23.9–25.4%) to 28.9% (95% CI: 28.1–29.7%), respectively, p<0.001). We identified the following independent specific determinants of CM use: gender, nationality, age, lifestyle and BMI. Female gender and nationality were the most specific determinants of CM use. Current smoking, being overweight and obesity were determinants of non-use of CM, while regular consumption of fruits and/or vegetables and regular physical activity were determinants of CM use.

**Data Availability Statement:** The data underlying the results presented in the study are available from the Swiss Federal Statistical Office (https://

www.bfs.admin.ch/bfs/fr/home/statistiques/sante/
enquetes/sgb.html#-1778047807).

**Funding:** The authors received no specific funding
for this work.

**Competing interests:** The authors have declared
that no competing interests exist.

## Conclusion

Prevalence of CM use significantly increased in Switzerland from 2012 to 2017. Gender, nationality, age, lifestyle and BMI were independent specific determinants of CM use as compared to conventional health care use. Healthier lifestyle was associated with CM use, which may have potentially significant implications for public health and preventive medicine initiatives. The nationality of CM users underlines the role of culture in driving the choice to use CM but also raises the question of whether all populations have equal access to CM within a same country.

## 1. Introduction

The use of complementary medicine (CM) increased considerably during the 1990s in many countries [1–11] and CM is now used by substantial portions of the general populations in a number of countries [9]. Based on the data from the 2014 European Social Survey, 25.9% of the general population in Europe had used CM during the last 12 months, varying from 10% in Hungary to almost 40% in Germany [12]. Outside the European Union, the 12-month prevalence of CM use was found to range from 9.8% to 76% [9]. These large variations in reported use are mainly due to the substantial heterogeneity of studies regarding the population characteristics, definitions of CM, time periods over which CM use was measured, as well as the methodology of studies [4, 13]. However, prevalence in each country is also influenced by economic, political and socio-cultural factors, including costs, accessibility of biomedical services, regulation of CM [9] and countries' respective health expenditure [4].

In a 2009 referendum, two-thirds of Swiss voters approved a new article in the Federal Constitution providing that CM should be recognized by the authorities. In 2017, four CM therapies (anthroposophic medicine, homeopathy, herbal medicine and traditional Chinese medicine (TCM)) were approved for full coverage by mandatory basic health insurance if delivered by a certified physician [14]. Private complementary insurances cover these therapies if delivered by a therapist, as well as other CM. The reimbursed CM therapies vary between private insurances. Based on the data from the Swiss Health Survey 2007 and 2012, 23.0% and 25.0%, respectively, of the population aged 15 and older had used at least one CM method in the previous 12 months [15, 16].

The most frequently used CM therapies in Europe are massage therapy, homeopathy, osteopathy, herbal medicine, acupuncture and chiropractic [12, 17]. The most frequently reported reason for CM use, as reflected in 84% of publications included in a worldwide systematic review, was the expected benefits of CM [18]. This includes treatment of illnesses, alleviation of symptoms, reduction of side effects of conventional medicine, maintenance of well-being, or prevention of disease. There is evidence that CM is frequently used as an adjunct to biomedical treatment by patients with serious disease such as cancers, or to self-manage long-term health complaints like lower back pain [4, 9, 17, 19–22]. Having an internal health locus of control was a frequently reported reason for CM use in Western populations [21]. Dissatisfaction with conventional medicine was reported in 37% of publications [18]. Furthermore, only 8% of CM users in Europe were found to use CM exclusively (alternative use), without any visits to medical professionals in the last 12 months [12]. This is in line with the increasing development of "medical pluralism" (i.e., the use of multiple forms of health care) [23].

Use of CM was shown to be associated with sociodemographic and health-related determinants. Sociodemographic determinants included female gender, being middle-aged, higher

levels of education, and income, while health-related determinants included poor self-reported health, chronic disease, and serious illness [4, 7, 9, 12, 15–17, 21, 24–30]. However, many existing CM studies did not investigate CM use according to the different CM therapies, which makes interpretation of the results difficult [21, 31, 32]. Furthermore, the above-mentioned determinants do not appear to be clearly specific to CM use and might be also associated with consultation of conventional health practitioners [32].

The aim of this study was to determine the prevalence of use of CM in the general population in Switzerland, as well as its development since the previous Swiss Health Survey in 2012, and to examine specific and non-specific sociodemographic, lifestyle and health-related determinants of CM use as compared to determinants of conventional health care use.

## 2. Methods

### 2.1. Data sources and study samples

The cross-sectional Swiss Health Survey has been conducted every five years since 1992 by the Swiss Federal Statistical Office (FSO) [33]. The survey is a sample drawn from all residents of Switzerland aged 15 years and above. It provides nationally representative information on health in the general population, including people's state of health, lifestyle, alcohol and drug abuse, physical exercise, health insurance and use of health services. The survey collects data using computer-assisted telephone (or face-to-face) interviews followed by self-completed written questionnaires. Questionnaires address questions which cannot be asked on the telephone (e.g., because the respondent needs to consult documents or due to the intimate nature of some questions).

We used data from the 2017 survey, the most recent national health survey available in Switzerland, and compared these data with those from 2012. In 2017, 43,769 persons were contacted, of which 22,134 (50.6%) participated in the telephone or face-to-face interview. The subsequent written questionnaires were returned by 18,832 (85.1%) of the 22,134 participants, resulting in a response rate of 43.0%. We restricted our analysis to the 18,832 participants who returned the questionnaire because information about CM use was only available in the written questionnaires. The 2012 sample has been described elsewhere [15].

The FSO provides anonymous data from the Swiss Health Survey upon request. The analysis of these data does not require approval of an ethics committee.

### 2.2. Definition of CM categories

In the written questionnaire, respondents were asked whether they had used the following therapies in the past 12 months: osteopathy, naturopathy, homeopathy, herbal medicine, acupuncture, shiatsu or reflexology, TCM, Ayurveda, or other therapies such as kinesiology, Feldenkrais, autogenic training, neural therapy, bioresonance therapy and anthroposophic medicine. We defined four CM categories: (1) TCM, including acupuncture; (2) homeopathy; (3) herbal medicine; (4) other CM therapies (shiatsu, reflexology, osteopathy, Ayurveda, naturopathy, kinesiology, Feldenkrais, autogenic training, neural therapy, bioresonance therapy, anthroposophic medicine). Each participant could be allocated to several CM categories.

### 2.3. Conventional health care use

Conventional health care use included consultations with general practitioners (GP) and other medical specialists who graduated from the university of medicine in Switzerland or who obtained a recognition of a foreign graduation. It does not include dentists. In order to avoid a

bias related to gender and the recommended annual check-up, visits to gynecologists were not considered.

## 2.4. Sociodemographic, lifestyle and health-related characteristics

Based on the data from each Swiss Health Survey, the FSO generates and validates a set of sociodemographic, lifestyle and health-related indicators [34]. In this study, we considered the following indicators:

- Sociodemographic indicators: age (15–24 as reference, 25–44, 45–64, $\geq$65 years old), gender (female as reference, male), educational level (primary education as reference, secondary education, tertiary education), marital status (single as reference, married, divorced/separated, widowed), housing occupancy status (renter as reference, owner, free housing i.e., paid by employer, relative, friend), occupation (economically inactive as reference, unemployed or homemaker, employed), nationality (Swiss as reference, northern/western European, southern European, eastern European, non-European), linguistic region (German-speaking Switzerland incl. Romansh-speaking Switzerland as reference, French-speaking Switzerland, Italian-speaking Switzerland), region of residence (urban region as reference, intermediate region, rural region).

- Physical health indicators: body mass index (BMI) (underweight, normal weight as reference, overweight, obese), physical disorder (i.e., back pain, feeling weak, stomach ache or abdominal pain, flatulence, diarrhea, constipation, insomnia, headache, heart irregularity, chest pain, fever) in the past 4 weeks (none as reference, moderate, severe), sleep disorder (none or few disorders as reference, moderate, pathological), long-lasting or chronic disease/condition ($\geq$ 6 past months). In addition to the aforementioned physical health indicators, we considered the following variables: pregnancy (no as reference, yes), allergies (no as reference, yes), cancer (no as reference, yes), intensity of headache or migraine in the past 4 weeks (none as reference, moderate, high).

- Mental health indicators: psychological distress during the past 4 weeks (low as reference, moderate, high), depression during the past 2 weeks (none or minimal as reference, slight, moderate, moderately severe, severe), impact of health concerns on lifestyle (living without thinking about health as reference, health concerns affect lifestyle, health concerns determine lifestyle).

- Lifestyle indicators: physical activity (none if moderate physical activity < 30 minutes per week or intensive physical activity < once a week as reference, partially active if 30–149 minutes of moderate physical activity per week or intensive physical activity once a week, sufficiently active if moderate physical activity $\geq$ 150 minutes per week or intensive physical activity twice a week, trained if intensive physical activity $\geq$ 3 times a week), fruit and/or vegetable consumption (< 5 days per week as reference, 0–2 portion per day $\geq$ 5 days per week, 3–4 portions per days $\geq$ 5 days per week, $\geq$ 5 portions per day $\geq$ 5 days per week), daily tobacco consumption (none as reference, occasional smoker, daily smoker), occasional drunkenness in the past 12 months (none in the past 12 months as reference, lifetime non-drinker or abstainer, < once a month, every month, $\geq$ once a week), last cannabis consumption (none as reference, >12 months, $\leq$ 12 months, $\leq$ 30 days).

- Personal resources and social support indicators: mastery (i.e., extent to which people see themselves as being in control of the forces that importantly affect their live) (low, moderate, high as reference), social supports (low as reference, moderate, high),

- Use of the health care system indicators: consultation with GP in the past 12 months (no as reference, yes), consultation with other medical specialists (except gynecologist) in the past 12 months, supplemental health insurance for CM (no as reference, yes, do not know).

### 2.5. Statistical analysis

Descriptive statistics for the categorical variables are presented as numbers and unweighted percentages, as well as weighted percentage and 95% confidence interval of the weighted percentage, using the FSO's survey weights. Comparisons between weighted data from the surveys from 2012 and 2017 were provided using the chi-squared test. Associations between CM use or conventional health care use and sociodemographic, lifestyle and health-related variables were determined in weighted bivariate analyses using a chi-squared test for categorical variables and a *t* test for continuous variables. Independent determinants of CM use and conventional health care use were assessed using multivariate weighted logistic regression models. Variables with a significance level below 0.10 in the bivariate analyses were included in the multivariate model. Only factors associated with the outcome variable of the multivariate regression with p<0.05 were kept in the final model. A backward selection procedure was applied using the likelihood-ratio test. All determinants with a significant likelihood-ratio test (p<0.05) were kept in the final model. Factors which, in the final models, were independent determinants of CM use but not conventional health care use were defined as independent specific determinants of CM use. Factors which were independent determinants of both CM use and conventional health care use in the final models were defined as independent non-specific determinants of CM use.

All tests were two-sided with a significance level of 0.05. Statistical analysis was performed using R, Version 4.1.0 [35].

## 3. Results

### 3.1. Prevalence of CM use and of conventional health care use

Table 1 shows the prevalence of CM use in the past 12 months in 2012 and 2017 according to the CM categories of the self-completed written questionnaire. Prevalence of CM use significantly increased between 2012 and 2017 (from 24.7% (95% CI: 23.9–25.4%) to 28.9% (95% CI: 28.1–29.7%), respectively, p<0.001). This significant increase concerns in particular osteopathy (p<0.001), naturopathy (p = 0.003), herbal medicine (p<0.001) and TCM excluding acupuncture (p = 0.02).

Table 2 shows the prevalence of conventional health care use in the past 12 months in 2012 and 2017. Both consultation with a GP and with other medical specialists significantly increased between 2012 and 2017 (GP: 66.6% (95% CI: 65.7–67.5%) versus 70.7% (95% CI: 69.9–71.5%), respectively, p<0.001; other medical specialists: 36.5% (95% CI: 35.6–37.4%) versus 43.1% (95% CI: 42.3–44.0%), respectively, p<0.001).

### 3.2. Supplemental health insurance for CM

Supplemental health insurance for CM significantly increased from 2012 to 2017 (Table 3).

### 3.3. Sociodemographic, lifestyle, and health-related determinants of CM and conventional health care use

S1 and S2 Tables provide a description and associations of sociodemographic, lifestyle and health-related characteristics with CM use and with conventional health care use. Independent

**Table 1. Prevalence of complementary medicine use in the past 12 months according to the self-completed questionnaire from the Swiss Health Survey 2012 and 2017.**

| | 2017 (N = 18,832[a]) | | 2012 (N = 18,357[a]) | | p-value |
|---|---|---|---|---|---|
| | N (unweighted %) | weighted % (95% CI) | N (unweighted %) | weighted % (95% CI) | |
| Any type of complementary medicine[b] | 5654 (30.3%) | 28.9% (28.1–29.7) | 5018 (27.5%) | 24.7% (23.9–25.4) | <0.001 |
| Osteopathy | 1930 (10.3%) | 9.5% (9.0–10.0) | 1459 (8.1%) | 6.8% (6.4–7.2) | <0.001 |
| Naturopathy | 1799 (9.6%) | 8.8% (8.3–9.0) | 1597 (8.8%) | 7.7% (7.2–8.2) | 0.003 |
| Homeopathy | 1731 (9.3%) | 8.4% (8.0–8.9) | 1662 (9.2%) | 8.2% (7.7–8.7) | 0.68 |
| Herbal medicine | 1369 (7.3%) | 7.0% (6.6–7.4) | 1014 (5.6%) | 5.0% (4.6–5.4) | <0.001 |
| Other therapies (kinesiology, Feldenkrais, autogenic training, neural therapy, bioresonance therapy, anthroposophic medicine) | 1323 (7.1%) | 6.9% (6.5–7.4) | 1242 (6.9%) | 6.1% (5.7–6.6) | 0.32 |
| Acupuncture | 1120 (6.0%) | 5.9% (5.5–6.3) | 1007 (5.6%) | 4.9% (4.5–5.3) | 0.06 |
| Shiatsu/reflexology | 884 (4.7%) | 4.5% (4.2–4.9) | 863 (4.8%) | 4.3% (4.0–4.7) | 0.95 |
| Traditional Chinese medicine (excluding acupuncture) | 472 (2.5%) | 2.5% (2.2–2.8) | 391 (2.2%) | 1.9% (1.7–2.2) | 0.02 |
| Ayurveda | 221 (1.2%) | 1.1% (1.0–1.3) | 202 (1.1%) | 0.9% (0.8–1.1) | 0.52 |

N, number; CI, confidence interval

[a]Representative sample of the general population > 15 years old in Switzerland

[b] Participants who used at least one complementary medicine therapy in the past 12 months.

determinants of TCM including acupuncture, homeopathy, herbal medicine, other CM therapies, CM non-user, consultation with GP and consultation with other medical specialists are shown in Tables 4–10, respectively.

**3.3.1. Sociodemographic determinants.** *Independent specific determinants of CM use as compared to independent determinants of conventional health care use*. Female gender was a strong independent determinant of CM use. Nationality of participants such as southern, eastern European and non-European was a strong independent determinant of non-use of CM. Age profiles differed according to CM category, while aging (≥65 years old) was an independent determinant of both consultation with GP and consultation with other medical specialists. Accordingly, employment was a determinant of CM use, whereas economic inactivity (including retired persons) was a determinant of conventional health care use. Young age was an independent determinant of homeopathy use. Being a home owner was an independent determinant of other CM therapies use. Living in an urban area was an independent determinant of consultation with other medical specialists.

*Non-specific determinants of CM use*. High level of education and living in the French-speaking part of Switzerland were independent determinants of both CM use and consultation with other medical specialists, but not of consultation with a GP.

**Table 2. Prevalence of conventional health care use in the past 12 months according to the telephone interviews from Swiss Health Survey 2012 and 2017.**

| | 2017 (N = 22,134[a]) | | 2012 (N = 21,597[a]) | | p-value |
|---|---|---|---|---|---|
| | N (unweighted %) | weighted % (95% CI) | N (unweighted %) | weighted % (95% CI) | |
| General practitioner | 15136 (71.5%) | 70.7% (69.9–71.5) | 14047 (67.5%) | 66.6% (65.7–67.5) | <0.001 |
| Other medical specialists | 9055 (42.8%) | 43.1% (42.3–44.0) | 7746 (37.2%) | 36.5% (35.6–37.4) | <0.001 |

N, number; CI, confidence interval

[a]Representative sample of the general population > 15 years old in Switzerland

**Table 3. Supplemental health insurance for complementary medicine (Swiss Health Survey 2012 and 2017).**

|  | 2017 (N = 18,832[a]) | | 2012 (N = 18,357[a]) | | p-value |
|---|---|---|---|---|---|
|  | N[b] (unweighted %) | weighted % (95% CI) | N (unweighted %) | weighted % (95% CI) | |
| Supplemental health insurance for complementary medicine |  |  |  |  | <0.001 |
| Yes | 10815 (57.8%) | 54.9% (54.1–55.8) | 9920 (54.5%) | 51.2% (50.2–52.1) | |
| No | 5600 (29.9%) | 32.0% (31.2–32.8) | 5877 (32.3%) | 34.2% (33.3–35.1) | |
| Don't know | 2292 (12.3%) | 13.1% (12.5–13.7) | 2396 (13.2%) | 14.6% (13.9–15.4) | |

N, number; CI, confidence interval

[a]Representative sample of the general population > 15 years old in Switzerland

[b]125 participants did not answer the question (missing data)

**Table 4. Determinants of traditional Chinese medicine use including multivariate logistic regression.**

|  | OR (95% CI) | p-value | p-value backward selection procedure (likelihood-ratio test) |
|---|---|---|---|
| Age, years |  |  | <0.001 |
| 15–24 | - |  |  |
| 25–44 | 1.61 (1.18–2.23) | 0.003 |  |
| 45–64 | 1.57 (1.15–2.14) | 0.005 |  |
| 65+ | 1.14 (0.78–1.66) | 0.50 |  |
| Gender, female | 1.93 (1.63–2.30) | <0.001 | <0.001 |
| Educational level |  |  | 0.04 |
| Primary education | - |  |  |
| Secondary education | 1.29 (0.99–1.69) | 0.05 |  |
| Tertiary education | 1.35 (1.01–1.80) | 0.04 |  |
| Occupation |  |  | 0.009 |
| Economically inactive | - |  |  |
| Unemployed/homemaker | 1.02 (0.51–2.02) | 0.96 |  |
| Employed | 1.05 (1.05–1.68) | 0.02 |  |
| Nationality |  |  | 0.006 |
| Swiss | - |  |  |
| Northern/western European | 0.88 (0.66–1.18) | 0.40 |  |
| Southern European | 0.73 (0.52–1.03) | 0.07 |  |
| Eastern European | 0.59 (0.39–0.90) | 0.01 |  |
| Non-European | 0.90 (0.41–1.97) | 0.79 |  |
| Linguistic region of Switzerland |  |  | 0.006 |
| German-speaking incl. Romansh-speaking | - |  |  |
| French-speaking | 1.27 (1.08–1.50) | 0.005 |  |
| Italian-speaking | 1.05 (0.79–1.39) | 0.76 |  |
| Body mass index |  |  | 0.01 |
| Underweight | 1.13 (0.78–1.63) | 0.53 |  |
| Normal weight | - |  |  |
| Overweight | 0.99 (0.83–1.18) | 0.90 |  |
| Obese | 0.70 (0.52–0.94) | 0.02 |  |
| Physical disorder in the past 4 weeks |  |  | <0.001 |
| None or few | - |  |  |
| Moderate | 1.40 (1.16–1.69) | <0.001 |  |
| Severe | 1.41 (1.15–1.74) | <0.001 |  |

*(Continued)*

**Table 4.** (Continued)

| | OR (95% CI) | p-value | p-value backward selection procedure (likelihood-ratio test) |
|---|---|---|---|
| Long-lasting or chronic disease/condition (≥ 6 past months) | 1.27 (1.08–1.50) | 0.004 | <0.001 |
| Daily tobacco consumption | | | <0.001 |
| None | - | | |
| Occasional smoker | 0.92 (0.70–1.22) | 0.57 | |
| Daily smoker | 0.71 (0.57–0.89) | 0.002 | |
| Consultation with general practitioner in the past 12 months | 1.64 (1.34–1.99) | <0.001 | <0.001 |
| Consultation with other medical specialist (except gynecologist) in the past 12 months | 1.64 (1.40–1.93) | <0.001 | <0.001 |

OR = odds ratio; CI = confidence interval

Full model included age, gender, education level, housing occupancy status, occupation, nationality, linguistic region, region of residency, body mass index, physical disorder in the past 4 weeks, sleep disorder, long-lasting or chronic disease/condition (≥ 6 past months), allergies, cancer, intensity of headache or migraine in the past 4 weeks, psychological distress in the past 4 weeks, depression in the past 2 weeks, impact of health concerns on lifestyle, fruit and/or vegetable consumption, daily tobacco consumption, last cannabis consumption, mastery, social support, consultation with general practitioner in the past 12 months, consultation with other medical specialists (except gynecologist) in the past 12 months, survey weights.

**Table 5. Determinants of homeopathy use using multivariate logistic regression.**

| | OR (95% CI) | p-value | p-value backward selection procedure (likelihood-ratio test) |
|---|---|---|---|
| Age, years | | | <0.001 |
| 15–24 | - | - | |
| 25–44 | 0.76 (0.60–0.96) | 0.02 | |
| 45–64 | 0.83 (0.66–1.05) | 0.12 | |
| 65+ | 0.48 (0.36–0.65) | <0.001 | |
| Gender, female | 2.03 (1.73–2.37) | <0.001 | <0.001 |
| Educational level | | | <0.001 |
| Primary education | - | | |
| Secondary education | 1.34 (1.06–1.70) | 0.01 | |
| Tertiary education | 1.82 (1.42–2.35) | <0.001 | |
| Nationality | | | <0.001 |
| Swiss | - | | |
| Northern/western European | 0.91 (0.68–1.21) | 0.51 | |
| South European | 0.57 (0.40–0.80) | 0.001 | |
| Eastern European | 0.32 (0.20–0.53) | <0.001 | |
| Non-European | 0.30 (0.13–0.67) | 0.003 | |
| Linguistic region | | | <0.001 |
| German-speaking incl. Romansh-speaking Switzerland | - | | |
| French-speaking Switzerland | 1.69 (1.46–1.96) | <0.001 | |
| Italian-speaking Switzerland | 1.22 (0.94–1.58) | 0.14 | |
| Physical disorder in the past 4 weeks | | | <0.001 |
| None or few | - | - | |
| Moderate | 1.25 (1.05–1.49) | 0.009 | |
| Severe | 1.57 (1.30–1.89) | <0.001 | |
| Allergies | 1.42 (1.22–1.65) | <0.001 | <0.001 |
| Impact of health concerns on lifestyle | | | <0.001 |
| Living without thinking about health | - | | |
| Health concerns affect lifestyle | 1.71 (1.32–2.21) | <0.001 | |

(Continued)

**Table 5.** (Continued)

| | OR (95% CI) | p-value | p-value backward selection procedure (likelihood-ratio test) |
|---|---|---|---|
| Health concerns determine lifestyle | 1.81 (1.35–2.43) | <0.001 | |
| Fruit and/or vegetable consumption | | | <0.001 |
| < 5 days/week | - | | |
| 0–2 portions/day, ≥5 days/week | 1.36 (0.96–1.92) | 0.08 | |
| 3–4 portions/day, ≥5 days/week | 1.64 (1.17–2.32) | 0.005 | |
| ≥5 portions/day, ≥5 days/week | 1.84 (1.30–2.63) | <0.001 | |
| Occasional drunkenness in the past 12 months | | | 0.009 |
| Lifetime non-drinker, abstainer | 0.75 (0.60–0.95) | 0.02 | |
| None in the past 12 months | - | - | |
| <1/month to every month | 0.91 (0.78–1.06) | 0.24 | |
| ≥ 1/week | 1.21 (0.80–1.84) | 0.37 | |
| Mastery | | | 0.002 |
| Low | 1.35 (1.11–1.64) | 0.002 | |
| Moderate | 1.15 (0.98–1.36) | 0.09 | |
| High | - | | |
| Social supports | | | 0.03 |
| Low | - | | |
| Moderate | 1.26 (0.94–1.69) | 0.12 | |
| High | 1.39 (1.03–1.86) | 0.03 | |
| Consultation with general practitioner in the past 12 months | 1.23 (1.05–1.45) | 0.009 | 0.002 |

OR = odds ratio; CI = confidence interval

Full model included age, gender, education level, marital status, housing occupancy status, occupation, nationality, linguistic region, region of residency, body mass index, physical disorder in the past 4 weeks, sleep disorder, long-lasting or chronic disease/condition (≥ 6 past months), allergies, cancer, intensity of headache or migraine in the past 4 weeks, psychological distress in the past 4 weeks, depression in the past 2 weeks, impact of health concerns on lifestyle, physical activity, fruit and/or vegetable consumption, daily tobacco consumption, occasional drunkenness in the past 12 months, last cannabis consumption, mastery, social support, consultation with general practitioner in the past 12 months, consultation with other medical specialists (except gynecologist) in the past 12 months, survey weights.

**3.3.2. Lifestyle determinants.** *Independent specific determinants of CM use as compared to independent determinants of conventional health care use*. Consumption of fruits and/or vegetables was an independent determinant of homeopathy, herbal medicine, and other CM therapies use. Daily tobacco consumers were significantly under-represented among TCM and other CM therapies users.

*Non-specific determinants of CM use*. Sufficient physical activity was an independent determinant of herbal medicine and other CM therapies use, whereas partially active participants were significantly under-represented among participants who consulted another medical specialist. Lifetime non-drinkers/abstinent participants were significantly under-represented among homeopathy users and among participants who consulted another medical specialist. Last consumption of cannabis in the past 30 days was an independent determinant of herbal medicine use. Consumption of cannabis in the past ≥30 days was an independent determinant of other CM therapies use and of consultation with other medical specialists.

**3.3.3. Health-related determinants.** *Independent specific determinants of CM use as compared to independent determinants of conventional health care use*. Participants that were overweight or obese were significantly under-represented among CM users, regardless of CM category, whereas it was an independent determinant of consultation with conventional physicians.

**Table 6. Determinants of herbal medicine use using multivariate logistic regression.**

| | OR (95% CI) | p-value | p-value backward selection procedure (likelihood-ratio test) |
|---|---|---|---|
| Age, years | | | <0.001 |
| 15–24 | - | - | |
| 25–44 | 1.24 (0.93–1.67) | 0.15 | |
| 45–64 | 1.53 (1.14–2.03) | 0.004 | |
| 65+ | 0.99 (0.70–1.40) | 0.95 | |
| Gender, female | 2.41 (2.01–2.89) | <0.001 | <0.001 |
| Educational level | | | <0.001 |
| Primary education | - | - | |
| Secondary education | 1.43 (1.10–1.87) | 0.008 | |
| Tertiary education | 1.82 (1.37–2.42) | <0.001 | |
| Nationality | | | <0.001 |
| Swiss | - | - | |
| Northern/western European | 0.97 (0.73–1.29) | 0.84 | |
| South European | 0.64 (0.44–0.92) | 0.02 | |
| Eastern European | 0.66 (0.43–1.01) | 0.05 | |
| Non-European | 0.32 (0.13–0.78) | 0.01 | |
| Linguistic region | | | <0.001 |
| German-speaking incl. Romansh-speaking Switzerland | - | - | |
| French-speaking Switzerland | 1.64 (1.40–1.92) | <0.001 | |
| Italian-speaking Switzerland | 1.03 (0.77–1.39) | 0.82 | |
| Body mass index | | | 0.048 |
| Underweight | 1.07 (0.74–1.55) | 0.72 | |
| Normal | - | - | |
| Overweight | 1.01 (0.85–1.21) | 0.91 | |
| Obese | 0.72 (0.53–0.97) | 0.03 | |
| Physical disorder in the past 4 weeks | | | <0.001 |
| None or few | - | - | |
| Moderate | 1.13 (0.93–1.37) | 0.22 | |
| Severe | 1.65 (1.32–2.05) | <0.001 | |
| Allergies | 1.23 (1.04–1.45) | 0.02 | 0.005 |
| Intensity of headache or migraine in the past 4 weeks | | | <0.001 |
| None | - | - | |
| Moderate | 0.99 (0.83–1.19) | 0.93 | |
| High | 0.57 (0.40–0.82) | 0.002 | |
| Impact of health concerns on lifestyle | | | <0.001 |
| Living without thinking about health | - | - | |
| Health concerns affect lifestyle | 1.98 (1.45–2.70) | <0.001 | |
| Health concerns determine lifestyle | 2.12 (1.51–2.98) | <0.001 | |
| Physical activity | | | 0.007 |
| None | - | - | |
| Partially active | 1.43 (0.96–2.14) | 0.08 | |
| Sufficiently active, trained | 1.63 (1.12–2.39) | 0.01 | |
| Fruit and/or vegetable consumption | | | <0.001 |
| < 5 days/week | - | - | |
| 0–2 portions/day, ≥5 days/week | 1.51 (1.05–2.16) | 0.02 | |
| 3–4 portions/day, ≥5 days/week | 1.62 (1.13–2.32) | 0.008 | |
| ≥5 portions/day, ≥5 days/week | 1.94 (1.35–2.79) | <0.001 | |

*(Continued)*

**Table 6.** (Continued)

|  | OR (95% CI) | p-value | p-value backward selection procedure (likelihood-ratio test) |
|---|---|---|---|
| Last cannabis consumption |  |  | 0.001 |
| None | - | - |  |
| >12 months | 1.11 (0.92–1.34) | 0.27 |  |
| ≤ 12 months | 1.25 (0.74–2.10) | 0.40 |  |
| ≤ 30 days | 1.92 (1.29–2.84) | 0.001 |  |
| Mastery |  |  | <0.001 |
| Low | 1.73 (1.41–2.12) | <0.001 |  |
| Moderate | 1.29 (1.07–1.56) | 0.007 |  |
| High | - | - |  |

OR = odds ratio; CI = confidence interval

Full model included age, gender, education level, marital status, occupation, nationality, linguistic region, body mass index, physical disorder in the past 4 weeks, sleep disorder, long-lasting or chronic disease/condition (≥ 6 past months), allergies, cancer, intensity of headache or migraine in the past 4 weeks, psychological distress in the past 4 weeks, depression in the past 2 weeks, impact of health concerns on lifestyle, physical activity, fruit and/or vegetable consumption, daily tobacco consumption, last cannabis consumption, mastery, consultation with general practitioner in the past 12 months, consultation with other medical specialists (except gynecologist) in the past 12 months, survey weights.

*Non-specific determinants of CM use.* Physical disorder was an independent determinant of both CM use and conventional health care use. Long-lasting or chronic disease/condition was an independent determinant of TCM and other CM therapies use, but not of homeopathy or herbal medicine use. It was a strong independent determinant of conventional health care use. Cancer was not associated with CM use, whereas it as was a strong independent determinant of consultation with other medical specialists. Having allergies was an independent determinant of homeopathy, herbal medicine, and other CM therapies use, as well as of consultation with other medical specialists. Except among TCM users, mastery and impact of health concerns on lifestyle were independent determinants of CM use and conventional health care use.

Consultation with GP and/or with other medical specialists was an independent determinant of CM use, except among herbal medicine users. Psychological distress was an independent determinant of consultation with GP, while depression was an independent determinant of consultation with other medical specialists.

## 4. Discussion

### 4.1. Key findings

Based on the data from the FSO's Swiss Health survey 2017, we observed that 28.9% of participants had used CM, 70.7% had visited a GP, and 43.1% had visited another medical specialist in the past 12 months. These findings showed a significant increase in the use of the health care system in Switzerland compared to 2012, with a parallel increase of participants having supplemental health insurance for CM.

Our findings showed distinct profiles of CM users as compared to conventional medicine users. Specific independent determinants of CM use are the following: gender, nationality, age, lifestyle, and BMI. Female gender and nationality were the most specific determinants of CM use. Nationality of participants such as southern or eastern European and non-European was a strong determinant of non-use of CM. Age below 65 years old was a determinant of CM use, with young people (15–24 years old) significantly over-represented among users of homeopathy. Moreover, current smoking, being overweight and obesity were determinants of non-use

**Table 7. Determinants of other CM therapies use using multivariate logistic regression.**

| | OR (95% CI) | p-value | p-value backward selection procedure (likelihood-ratio test) |
|---|---|---|---|
| Age, years | | | <0.001 |
| 15–24 | - | - | |
| 25–44 | 1.38 (1.11–1.70) | 0.003 | |
| 45–64 | 1.29 (1.04–1.61) | 0.02 | |
| 65+ | 0.92 (0.71–1.20) | 0.55 | |
| Gender, female | 2.16 (1.93–2.41) | <0.001 | <0.001 |
| Educational level | | | <0.001 |
| Primary education | - | - | |
| Secondary education | 1.39 (1.17–1.66) | <0.001 | |
| Tertiary education | 1.74 (1.43–2.10) | <0.001 | |
| Marital status | | | 0.02 |
| Single | - | - | |
| Married | 1.06 (0.92–1.22) | 0.42 | |
| Divorced/separated | 1.26 (1.03–1.54) | 0.03 | |
| Widowed | 1.39 (1.00–1.93) | 0.05 | |
| Housing occupancy status | | | <0.001 |
| Renter | - | - | |
| Owner | 1.30 (1.17–1.45) | <0.001 | |
| Free housing (paid by employer, relative, friend) | 0.87 (0.51–1.46) | 0.59 | |
| Occupation | | | <0.001 |
| Inactive | - | - | |
| Unemployed/housework | 1.14 (0.76–1.71) | 0.54 | |
| Employed | 1.28 (1.10–1.48) | 0.001 | |
| Nationality | | | <0.001 |
| Swiss | - | - | |
| Northern/western European | 0.83 (0.67–1.03) | 0.10 | |
| South European | 0.49 (0.38–0.62) | <0.001 | |
| Eastern European | 0.44 (0.33–0.58) | <0.001 | |
| Non-European | 0.32 (0.17–0.59) | <0.001 | |
| Linguistic region of Switzerland | | | <0.001 |
| German-speaking incl. Romansh-speaking | - | - | |
| French-speaking | 2.09 (1.87–2.33) | <0.001 | |
| Italian-speaking | 0.72 (0.58–0.89) | 0.003 | |
| Body mass index | | | <0.001 |
| Underweight | 1.04 (0.80–1.35) | 0.79 | |
| Normal | - | - | |
| Overweight | 0.87 (0.77–0.98) | 0.02 | |
| Obese | 0.77 (0.64–0.92) | 0.005 | |
| Physical disorder in the past 4 weeks | | | <0.001 |
| None or few | - | - | |
| Moderate | 1.28 (1.13–1.43) | <0.001 | |
| Severe | 1.62 (1.41–1.86) | <0.001 | |
| Long-lasting or chronic disease/condition ($\geq$ 6 past months) | 1.22 (1.09–1.37) | <0.001 | <0.001 |
| Allergies | 1.16 (1.04–1.30) | 0.009 | 0.002 |
| Impact of health concerns on lifestyle | | | <0.001 |
| Living without thinking about health | - | - | |

(*Continued*)

**Table 7.** (Continued)

| | OR (95% CI) | p-value | p-value backward selection procedure (likelihood-ratio test) |
|---|---|---|---|
| Health concerns affect lifestyle | 1.52 (1.28–1.81) | <0.001 | |
| Health concerns determine lifestyle | 1.75 (1.43–2.14) | <0.001 | |
| Physical activity | | | 0.004 |
| None | - | - | |
| Partially active | 1.28 (0.99–1.67) | 0.06 | |
| Sufficiently active, trained | 1.39 (1.08–1.78) | 0.01 | |
| Fruit and/or vegetable consumption | | | 0.003 |
| < 5 days/week | - | - | |
| 0–2 portions/day, ≥5 days/week | 1.14 (0.93–1.40) | 0.21 | |
| 3–4 portions/day, ≥5 days/week | 1.23 (1.00–1.51) | 0.05 | |
| ≥5 portions/day, ≥5 days/week | 1.35 (1.09–1.67) | 0.006 | |
| Daily tobacco consumption | | | <0.001 |
| None | - | - | |
| Occasional smoker | 0.99 (0.83–1.19) | 0.94 | |
| Daily smoker | 0.74 (0.64–0.85) | <0.001 | |
| Last cannabis consumption | | | <0.001 |
| None | - | - | |
| >12 months | 1.26 (1.11–1.43) | <0.001 | |
| ≤ 12 months | 1.53 (1.12–2.08) | 0.007 | |
| ≤ 30 days | 1.20 (0.87–1.65) | 0.27 | |
| Mastery | | | <0.001 |
| Low | 1.28 (1.11–1.47) | <0.001 | |
| Moderate | 1.06 (0.94–1.18) | 0.38 | |
| High | - | - | |
| Social supports | | | <0.001 |
| Low | - | - | |
| Moderate | 1.14 (0.93–1.41) | 0.21 | |
| High | 1.37 (1.11–1.69) | 0.003 | |
| Consultation with general practitioner in the past 12 months | 1.23 (1.09–1.38) | <0.001 | <0.001 |
| Consultation with other medical specialists (except gynecologist) in the past 12 months | 1.39 (1.25–1.54) | <0.001 | <0.001 |

OR = odds ratio; CI = confidence interval

Other complementary medicine therapies include shiatsu, reflexology, osteopathy, Ayurveda, naturopathy, kinesiology, Feldenkrais, autogenic training, neural therapy, bioresonance therapy, anthroposophic medicine.

Full model included age, gender, education level, marital status, housing occupancy status, occupation, nationality, linguistic region, region of residency, body mass index, physical disorder in the past 4 weeks, sleep disorder, long-lasting or chronic disease/condition (≥ 6 past months), allergies, cancer, intensity of headache or migraine in the past 4 weeks, psychological distress in the past 4 weeks, depression in the past 2 weeks, impact of health concerns on lifestyle, physical activity, fruit and/or vegetable consumption, daily tobacco consumption, occasional drunkenness in the past 12 months, last cannabis consumption, mastery, social support, consultation with general practitioner in the past 12 months, consultation with other medical specialists (except gynecologist) in the past 12 months, survey weights.

of CM, while regular consumption of fruits and/or vegetables and regular physical activity were determinants of CM use.

In addition, we observed specific profiles according to CM categories. Herbal medicine users were mainly healthy females reporting significantly more physical disorder and allergies than non-users of herbal medicine but without significant over-representation of chronic disease or conventional health care use. This raises the hypothesis that herbal medicine in Switzerland might be not only used to treat diseases but also for health promotion rather than to

**Table 8. Determinants of CM non-user using multivariate logistic regression.**

| | OR (95% CI) | p-value | p-value backward selection procedure (likelihood-ratio test) |
|---|---|---|---|
| Age, years | | | <0.001 |
| 15–24 | - | - | |
| 25–44 | 0.74 (0.62–0.88) | <0.001 | |
| 45–64 | 0.76 (0.64–0.90) | 0.001 | |
| 65+ | 1.04 (0.84–1.29) | 0.69 | |
| Gender, female | 0.46 (0.42–0.51) | <0.001 | <0.001 |
| Educational level | | | <0.001 |
| Primary education | - | - | |
| Secondary education | 0.74 (0.64–0.87) | <0.001 | |
| Tertiary education | 0.62 (0.52–0.74) | <0.001 | |
| Housing occupancy status | | | |
| Renter | - | - | <0.001 |
| Owner | 0.80 (0.73–0.88) | <0.001 | |
| Free housing (paid by employer, relative, friend) | 1.09 (0.67–1.78) | 0.73 | |
| Occupation | | | <0.001 |
| Inactive | - | - | |
| Unemployed/housework | 0.93 (0.65–1.34) | 0.71 | |
| Employed | 0.78 (0.68–0.89) | <0.001 | |
| Nationality | | | <0.001 |
| Swiss | - | - | |
| Northern/western European | 1.16 (0.95–1.42) | 0.13 | |
| South European | 1.75 (1.43–2.14) | <0.001 | |
| Eastern European | 2.19 (1.71–2.80) | <0.001 | |
| Non-European | 2.38 (1.47–3.85) | <0.001 | |
| Linguistic region | | | <0.001 |
| German-speaking incl. Romansh-speaking Switzerland | - | - | |
| French-speaking Switzerland | 0.50 (0.45–0.55) | <0.001 | |
| Italian-speaking Switzerland | 0.97 (0.81–1.17) | 0.78 | |
| Body mass index | | | <0.001 |
| Underweight | 0.95 (0.75–1.21) | 0.68 | |
| Normal | - | - | |
| Overweight | 1.17 (1.05–1.30) | 0.006 | |
| Obese | 1.37 (1.16–1.62) | <0.001 | |
| Physical disorder in the past 4 weeks | | | <0.001 |
| None or few | - | - | |
| Moderate | 0.81 (0.73–0.90) | <0.001 | |
| Severe | 0.63 (0.56–0.72) | <0.001 | |
| Long-lasting or chronic disease/condition (≥ 6 past months) | 0.82 (0.73–0.92) | <0.001 | <0.001 |
| Allergies | 0.86 (0.78–0.96) | 0.004 | <0.001 |
| Impact of health concerns on lifestyle | | | <0.001 |
| Living without thinking about health | - | - | |
| Health concerns affect lifestyle | 0.68 (0.58–0.79) | <0.001 | |
| Health concerns determine lifestyle | 0.62 (0.52–0.75) | <0.001 | |
| Physical activity | | | 0.006 |
| None | - | - | |
| Partially active | 0.82 (0.65;1.04) | 0.10 | |

*(Continued)*

**Table 8.** (Continued)

| | OR (95% CI) | p-value | p-value backward selection procedure (likelihood-ratio test) |
|---|---|---|---|
| Sufficiently active, trained | 0.76 (0.61;0.95) | 0.01 | |
| Fruit and/or vegetable consumption | | | <0.001 |
| < 5 days/week | - | - | |
| 0–2 portions/day, ≥5 days/week | 0.80 (0.67–0.97) | 0.02 | |
| 3–4 portions/day, ≥5 days/week | 0.73 (0.61–0.88) | 0.001 | |
| ≥5 portions/day, ≥5 days/week | 0.69 (0.57–0.84) | <0.001 | |
| Daily tobacco consumption | | | <0.001 |
| None | - | - | |
| Occasional smoker | 1.03 (0.87–1.23) | 0.72 | |
| Daily smoker | 1.44 (1.26–1.64) | <0.001 | |
| Last cannabis consumption | | | <0.001 |
| None | - | - | |
| >12 months | 0.79 (0.70–0.89) | <0.001 | |
| ≤ 12 months | 0.69 (0.51–0.93) | 0.01 | |
| ≤ 30 days | 0.65 (0.49–0.87) | 0.004 | |
| Mastery | | | |
| Low | 0.74 (0.65–0.85) | <0.001 | <0.001 |
| Moderate | 0.89 (0.80–0.99) | 0.03 | |
| High | - | - | |
| Social support | | | 0.001 |
| Low | - | - | |
| Moderate | 0.91 (0.75–1.09) | 0.30 | |
| High | 0.79 (0.66–0.96) | 0.02 | |
| Consultation with general practitioner in the past 12 months | 0.79 (0.71–0.88) | <0.001 | <0.001 |
| Consultation with other medical specialists (except gynecologist) in the past 12 months | 0.71 (0.65–0.79) | <0.001 | <0.001 |

OR = odds ratio; CI = confidence interval

Full model included age, gender, education level, marital status, housing occupancy status, occupation, nationality, linguistic region, region of residency, body mass index, physical disorder in the past 4 weeks, sleep disorder, long-lasting or chronic disease/condition (≥ 6 past months), allergies, psychological distress in the past 4 weeks, depression in the past 2 weeks, impact of health concerns on lifestyle, physical activity, fruit and/or vegetable consumption, daily tobacco consumption, occasional drunkenness in the past 12 months, last cannabis consumption, mastery, social support, consultation with general practitioner in the past 12 months, consultation with other medical specialists (except gynecologist) in the past 12 months, survey weights.

treat illness. Homeopathy users were significantly over-represented by young people (15–24 years old). Long-lasting or chronic disease/condition was a determinant of TCM use and other CM therapies use.

## 4.2. Comparison of the study results to other studies

Prevalence of CM use and of consultation with conventional physicians observed in our study are similar to prevalence of use in Europe: in 2014, 28.9% in Switzerland versus 25.9% in Europe had used CM, 70.7% in Switzerland versus 76.3% in Europe had visited a GP, and 43.1% in Switzerland versus 44.6% in Europe had visited a medical specialist [12]. More specifically, prevalence of CM use in Switzerland mirrors in particular the use patterns of German-speaking countries (Austria, Germany) and northern countries (Denmark, Finland, Sweden, Estonia, Lithuania). These countries presented with the highest rates of prevalence in Europe.

**Table 9. Determinants of consultation with general practitioner using multivariate logistic regression.**

| | OR (95% CI) | p-value | p-value backward selection procedure (likelihood-ratio test) |
|---|---|---|---|
| Age, years | | | <0.001 |
| 15–24 | - | - | |
| 25–44 | 0.68 (0.58–0.80) | <0.001 | |
| 45–64 | 0.83 (0.71–0.97) | 0.02 | |
| 65+ | 1.50 (1.22–1.85) | <0.001 | |
| Educational level | | | 0.005 |
| Primary education | - | - | |
| Secondary education | 1.00 (0.87–1.15) | 0.99 | |
| Tertiary education | 0.88 (0.75–1.02) | 0.09 | |
| Occupation | | | 0.02 |
| Inactive | - | - | |
| Unemployed/housework | 0.90 (0.65–1.26) | 0.54 | |
| Employed | 0.86 (0.75–0.98) | 0.02 | |
| Body mass index | | | <0.001 |
| Underweight | 0.89 (0.70–1.13) | 0.36 | |
| Normal | - | - | |
| Overweight | 1.18 (1.06–1.31) | 0.002 | |
| Obese | 1.51 (1.28–1.79) | <0.001 | |
| Physical disorder in the past 4 weeks | | | <0.001 |
| None or few | - | - | |
| Moderate | 1.43 (1.29–1.58) | <0.001 | |
| Severe | 1.88 (1.63–2.16) | <0.001 | |
| Long-lasting or chronic disease/condition (≥ 6 past months) | 2.57 (2.29–2.88) | <0.001 | <0.001 |
| Depression in the past 2 weeks | | | 0.003 |
| None or minimal | - | - | |
| Slight | 1.15 (1.03–1.29) | 0.02 | |
| Moderate | 1.42 (0.95–2.12) | 0.08 | |
| Impact of health concerns on lifestyle | | | <0.001 |
| Living without thinking about health | - | - | |
| Health concerns affect lifestyle | 1.26 (1.10–1.44) | <0.001 | |
| Health concerns determine lifestyle | 1.43 (1.21–1.69) | <0.001 | |
| Mastery | | | 0.03 |
| Low | 1.16 (1.01–1.33) | 0.04 | |
| Moderate | 1.09 (0.98–1.20) | 0.12 | |
| High | - | - | |

OR = odds ratio; CI = confidence interval

Full model included age, gender, education level, marital status, housing occupancy status, occupation, linguistic region, body mass index, physical disorder in the past 4 weeks, sleep disorder, long-lasting or chronic disease/condition (≥ 6 past months), allergies, cancer, intensity of headache or migraine in the past 4 weeks, psychological distress in the past 4 weeks, depression in the past 2 weeks, impact of health concerns on lifestyle, physical activity, daily tobacco consumption, occasional drunkenness in the past 12 months, last cannabis consumption, mastery, social support, survey weights.

These findings indicate that prevalence of CM use might be influenced by cultural factors; underline the role of the German language in the diffusion of usage of CM since several principles of CM originated in Germany; and suggest the role of regulations in terms of inclusion of CM in biomedical practice and health insurance [4, 9, 12]. Indeed, in Switzerland, the mandatory basic health insurance covers anthroposophic medicine, homeopathy, herbal medicine

**Table 10. Determinants of consultation with other medical specialists using multivariate logistic regression.**

| | OR (95% CI) | p-value | p-value backward selection procedure (likelihood-ratio test) |
|---|---|---|---|
| Age, years | | | <0.001 |
| 15–24 | - | - | |
| 25–44 | 0.90 (0.77–1.05) | 0.18 | |
| 45–64 | 1.15 (0.99–1.35) | 0.08 | |
| 65+ | 1.51 (1.25–1.83) | <0.001 | |
| Educational level | | | <0.001 |
| Primary education | - | - | |
| Secondary education | 1.17 (1.03–1.35) | 0.02 | |
| Tertiary education | 1.37 (1.18–1.59) | <0.001 | |
| Occupation | | | 0.001 |
| Inactive | - | - | |
| Unemployed/housework | 0.76 (0.55–1.04) | 0.08 | |
| Employed | 0.83 (0.73–0.94) | 0.003 | |
| Linguistic region of Switzerland | | | <0.001 |
| German-speaking incl. Romansh-speaking | - | - | |
| French-speaking | 1.23 (1.12–1.36) | <0.001 | |
| Italian-speaking | 1.07 (0.92–1.25) | 0.37 | |
| Region of residence | | | 0.006 |
| Urban region | - | - | |
| Intermediate region | 0.93 (0.83–1.03) | 0.15 | |
| Rural region | 0.86 (0.77–0.96) | 0.008 | |
| Body mass index | | | 0.001 |
| Underweight | 1.02 (0.79–1.31) | 0.87 | |
| Normal | - | - | |
| Overweight | 1.05 (0.96–1.16) | 0.28 | |
| Obese | 1.28 (1.10–1.48) | 0.001 | |
| Physical disorder in the past 4 weeks | | | <0.001 |
| None or few | - | - | |
| Moderate | 1.31 (1.19–1.44) | <0.001 | |
| Severe | 1.57 (1.39–1.78) | <0.001 | |
| Long-lasting or chronic disease/condition ($\geq$ 6 past months) | 2.51 (2.28–2.76) | <0.001 | <0.001 |
| Allergies | 1.20 (1.09–1.33) | <0.001 | <0.001 |
| Cancer | 4.84 (3.03–7.73) | <0.001 | <0.001 |
| Psychological distress in the past 4 weeks | | | <0.001 |
| Low | - | - | |
| Moderate | 1.24 (1.07–1.43) | 0.005 | |
| High | 1.84 (1.42–2.39) | <0.001 | |
| Impact of health concerns on lifestyle | | | <0.001 |
| Living without thinking about health | - | - | |
| Health concerns affect lifestyle | 1.33 (1.16–1.53) | <0.001 | |
| Health concerns determine lifestyle | 1.33 (1.13–1.56) | <0.001 | |
| Physical activity | | | 0.003 |
| None | - | - | |
| Partially active | 0.79 (0.64–0.96) | 0.02 | |
| Sufficiently active, trained | 0.91 (0.76–1.09) | 0.31 | |
| Last cannabis consumption | | | <0.001 |
| None | - | - | |

(*Continued*)

**Table 10.** (Continued)

| | OR (95% CI) | p-value | p-value backward selection procedure (likelihood-ratio test) |
|---|---|---|---|
| >12 months | 1.11 (1.00–1.24) | 0.06 | |
| ≤ 12 months | 1.36 (1.04–1.77) | 0.02 | |
| ≤ 30 days | 1.31 (0.99–1.72) | 0.06 | |
| Occasional drunkenness in the past 12 months | | | <0.001 |
| Lifetime non-drinker, abstainer | 0.85 (0.75–0.97) | 0.02 | |
| None in the past 12 months | - | - | |
| <1/month to every month | 1.04 (0.95–1.15) | 0.40 | |
| ≥ 1/week | 0.84 (0.64–1.09) | 0.19 | |
| Mastery | | | 0.003 |
| Low | 1.19 (1.05–1.24) | 0.006 | |
| Moderate | 1.09 (0.99–1.20) | 0.07 | |
| High | - | - | |

OR = odds ratio; CI = confidence interval

Full model included age, gender, education level, marital status, occupation, nationality, linguistic region, region of residency, body mass index, physical disorder in the past 4 weeks, sleep disorder, long-lasting or chronic disease/condition (≥ 6 past months), allergies, cancer, intensity of headache or migraine in the past 4 weeks, psychological distress in the past 4 weeks, depression in the past 2 weeks, impact of health concerns on lifestyle, physical activity, daily tobacco consumption, occasional drunkenness in the past 12 months, last cannabis consumption, mastery, social support, survey weights.

and TCM delivered by a certified physician, and private supplemental health insurances including various conditions for reimbursement cover many CM. The impact of the reimbursement of some CM therapies by mandatory basic health insurance on CM use was not possible to differentiate in the questionnaire if the user resorted to a CM reimbursed by mandatory or private health insurance.

In accordance with most studies investigating the determinants of CM use, we found that sociodemographic determinants such as female gender, being middle-aged, and higher levels of education were associated with CM use [4, 7, 9, 12, 15–17, 21, 24–30].

Furthermore, our results show that CM users reported healthier lifestyles compared with non-users. They reported being physically active, being non-smokers and meeting national recommendations for intake of fruits and vegetables. These findings are in accordance with the profile of Australian consumers of CM [36] and with the findings of the National Health Interview Survey 2012 in the United States [37] in which CM users reported being motivated by CM to make positive health behavior changes, including exercising more regularly, eating healthier and reducing/stopping smoking or alcohol consumption.

In line with a recent systematic review assessing the predictive factors of complementary and alternative medicine use in the general population in Europe, we observed the self-report of a chronic disease to be associated with consulting a CM practitioner (TCM including acupuncture or other CM therapies), and to be non-specific determinants of CM use [32]. In contrast, we found that homeopathy and herbal medicine users did not report more chronic disease than non-users, which is in accordance with the findings from the European Social Survey 2014 in which herbal medicine was more often employed to improve quality of life and the use of homeopathy was not associated with any specific health problems [12].

### 4.3. Strengths and limitations of the study

We used data from the Swiss Health survey, a population-based design involving a large random sample of Switzerland's population. Detailed information on participants was available,

which allowed to determine profiles of both CM users and conventional medicine users. Additional strengths of the present study include the short timeframe of questions to reduce recall bias.

One limitation of the study is the absence of standardized CM categories in the questionnaire of the survey. In particular, the questionnaire included the following CM as a single category, whereas they probably cover different user profiles: kinesiology, Feldenkrais, autogenic training, neural therapy, bioresonance therapy, anthroposophic medicine. Additional limitations were the absence of information on frequency of CM use (single versus more frequent usage), as well as the absence of information on motivations for CM use (e.g., medical need, prevention and wellness promotion, cultural relevance). To address the latter limitation, we included in the analyses detailed information on patient-reported health status, assuming that people with a poor health status use CM to treat illness rather than for health promotion.

### 4.4. Relevance of the study and implications for policymakers

This study provides evidence that a healthier lifestyle is associated with CM use. However, as causation cannot be determined as part of this cross-sectional survey, it remains unclear whether CM use motivates behavior change, or whether being predisposed to make health behavior changes drives the choice to use CM. If CM can help improve patients' health behavior, this may have potentially significant implications for public health and preventive medicine initiatives, which thus warrants further research attention.

This study reveals that southern European, eastern European, and non-European are strongly under-represented among CM users. Accessibility to CM and factors limiting CM use in these populations should be assessed.

### 4.5. Conclusions

This study shows that prevalence of CM use significantly increased from 2012 to 2017 in Switzerland. Gender, nationality, age, lifestyle, and BMI were independent specific determinants of CM use as compared to conventional health care use. Although based on the data of this survey it could not be clarified whether CM use motivates behavior change or whether being predisposed to make health behavior changes drives the choice to use CM, it is worthwhile to consider that healthier lifestyle is associated with CM use. This may have potentially significant implications for public health and preventive medicine initiatives, and thus warrants further research attention. Finally, this study points out the role of nationality in the profile of CM users. This underlines the role of culture in driving the choice to use CM, but also raises the question of whether all populations have equal access to CM within a same country.

### Supporting information

**S1 Table. Sociodemographic and health-related characteristics of responders according to CM category.**
(DOCX)

**S2 Table. Sociodemographic and health-related characteristics of responders according to CM use and conventional health care use.**
(DOCX)

**S1 File.**
(PDF)

## Acknowledgments

We thank the Swiss Federal Statistical Office for making the data available. We thank Rachel Scholkmann for proofreading.

## Author Contributions

**Formal analysis:** Delphine Meier-Girard.

**Supervision:** Pierre-Yves Rodondi, Ursula Wolf.

**Writing – original draft:** Delphine Meier-Girard.

**Writing – review & editing:** Delphine Meier-Girard, Emmanuelle Lüthi, Pierre-Yves Rodondi, Ursula Wolf.

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
