## [Decision Letter · Decision Letter 0]

10 May 2022

PONE-D-22-07999Specific and non-specific determinants of use of complementary medicine in Switzerland: data from the 2017 Swiss Health SurveyPLOS ONE

Dear Dr. Meier-Girard,

Thank you for submitting your manuscript to PLOS ONE. After careful consideration, we feel that it has merit but does not fully meet PLOS ONE’s publication criteria as it currently stands. Therefore, we invite you to submit a revised version of the manuscript that addresses the points raised during the review process.

We look forward to receiving your revised manuscript.

Kind regards,

Jenny Wilkinson, PhD

Academic Editor

PLOS ONE

Journal Requirements:

Additional Editor Comments:

Thank you for your submission. Reviewer comments are provided to provide you the opportunity to address areas of your work that can be strengthened. In particular both reviewers have highlighted questions about presentation of the data and analytical decisions (e.g. use of non-standard age categories).

Reviewers' comments:

Reviewer's Responses to Questions

**Comments to the Author**

1. Is the manuscript technically sound, and do the data support the conclusions?

Reviewer #1: Yes

Reviewer #2: Partly

2. Has the statistical analysis been performed appropriately and rigorously? 

Reviewer #1: Yes

Reviewer #2: Yes

3. Have the authors made all data underlying the findings in their manuscript fully available?

Reviewer #1: Yes

Reviewer #2: No

4. Is the manuscript presented in an intelligible fashion and written in standard English?

Reviewer #1: Yes

Reviewer #2: Yes

5. Review Comments to the Author

Reviewer #1: The manuscript is well written and is very information and of interest to the readers. However, there are some clarifications and suggestions for the authors:

1. Primary/ important important objective of the study was to see the prevalence of CM and also it's comparison with previous data, so in the title of manuscript the mention of prevalence would be more apt.

2. The flow diagram similar to PRISMA flow chart: http://prisma-statement.org/prismastatement/flowdiagram.aspx

would have made the data inclusion more clear.

In table 1:

for " Any type of complementary medicine" the N is 5654 and therefore all other rows following this {Osteopathy 1930

Naturopathy (1799), Homeopathy (1731), Herbal medicine (1369), Other therapies (1323),Acupuncture (1120) ,

Shiatsu/reflexology (8840),Traditional Chinese medicine (472) and Ayurveda 221} should total to 5654 but it is not. Please clarify why

In table 3:

Supplemental health insurance for complementary medicine: Yes (10815), No (5600) and Don’t know(2292) should total to N= 18,832 but it is not. Please clarify why

Another important aspect is the duration of use of CM, some individuals use it for short duration for example for some short duration issue like 1- 2 days(constipation, diarrhoea, pain ) for 2-3 days. Was there any criteria for inclusion of data based on duration of use of CM.

The use of CM it is most of the time not taken after consultation from GP or other expert but is usually taken through recommendation by family members, friends or other acquaintances. Highlighting these sources other than experts would have been interesting as it may have safety issues

Reviewer #2: Thank you for the opportunity to review this work. The 2017 and 2012 survey data were large but may be outdated. The authors should be applauded for the effort to conduct add-on questionnaire with a relatively good response rate. Nonetheless, the ‘85.1%’ was based on the 18,832 as the numerator and the 22,134 (but not 43,769) as the denominator. Please clarify and add an overview of the Swiss Health Survey in relationship with this particular study.

The coverage by mandatory basic health insurance seems to be intuitively the main factor of the increased CM use. As such, the analysis and/or discussion should differentiate the marginal effect of individual preferences, given the insurance coverage. When was the CM covered (before 2012 / between 2012-2017 / after 2017)? Table 3 presented the ‘Supplemental health insurance for complementary medicine’ as a separate finding. Actually, this point could be relatively easy to address by analyzing secondary annual claim data that should be available in the developed high-income country context like Switzerland. This suggested approach not only could provide the ‘revealed preferences’ of Swiss individuals, but also reduce the unreliable operational definitions of the CM and conventional therapies perceived by the respondents as pointed out below.

Line 110-114 presented a series of therapies without clear definitions to the readers (and to the respondents). Given the fact that these therapies have been covered by the mandatory basic health insurance, they should be clearly defined, along with whether each of the therapies is fully reimbursable and whether out-of-pocket or copayment is required.

Line 114-118: What are the additional benefits of the arbitrary and non-standard CM categorization?

Line 120-123: Conventional health care is poorly defined and, therefore, could not be referred to as a control group for the research question of this study.

Line 130: Was the age variable collected as a categorical variable as presented here? If not so, how do the authors justify these arbitrary categories? Please clarify this point with the other ‘ever-continuous-but-now-categorical’ variables as well.

Line 163: Is ‘last cannabis consumption’ part of the Lifestyle indicators of the Swiss Health Survey? If not, please differentiate which variables are from those that are not.

6. PLOS authors have the option to publish the peer review history of their article (what does this mean?). If published, this will include your full peer review and any attached files.

Reviewer #1: **Yes: **Rimple Jeet Kaur

Reviewer #2: **Yes: **Assoc. Prof. Dr. Krit Pongpirul, MD, MPH, PhD.

---

## [Author Response · Author response to Decision Letter 0]

28 Jun 2022

We thank the reviewers for the careful reading of our manuscript and their valuable comments. We hope we could address their concerns and clarify the ambiguities.

Reviewer #1 

1. The manuscript is well written and is very information and of interest to the readers. However, there are some clarifications and suggestions for the authors:

1. Primary/ important important objective of the study was to see the prevalence of CM and also it's comparison with previous data, so in the title of manuscript the mention of prevalence would be more apt.

Prevalence of CM was namely an important objective of the study and we state it in the manuscript accordingly. Although we aimed to underline the originality of the study, namely the investigation of the specific and the non-specific determinants of the use of complementary medicine in Switzerland, we agree with the reviewer to add the term prevalence in the title. The new title reads now: Prevalence, specific and non-specific determinants of complementary medicine use in Switzerland: data from the 2017 Swiss Health Survey.

2. The flow diagram similar to PRISMA flow chart: would have made the data inclusion more clear.

The data inclusion is described in lines 100 to 105. We included all participants who returned the questionnaire given that information about CM use was only available in the written questionnaire (i.e. 18,882 participants out of 22,134). Therefore, we do not feel that this information would be better depicted by a flow diagram as the data come from a national survey. Other studies based the same database also did not present a flowchart (e.g. https://pubmed.ncbi.nlm.nih.gov/33563620/;
https://www.sciencedirect.com/science/article/pii/S221133552200122X). We hope that the reviewer agrees with our reasoning.

3.In table 1: for " Any type of complementary medicine" the N is 5654 and therefore all other rows following this {Osteopathy 1930, Naturopathy (1799), Homeopathy (1731), Herbal medicine (1369), Other therapies (1323),Acupuncture (1120) , Shiatsu/reflexology (8840),Traditional Chinese medicine (472) and Ayurveda 221} should total to 5654 but it is not. Please clarify why

Any type of complementary medicine is defined as participants who used osteopathy and/or naturopathy and/or homeopathy and/or herbal medicine and/or acupuncture and/or Shiatsu/reflexology /or Traditional Chinese medicine and/or Ayurveda /or other therapies in the past 12 months.

Therefore, each participant could have been allocated to several CM therapies. For example, if a participant used osteopathy and acupuncture, he or she was accounted in the category osteopathy and in the category acupuncture, respectively but once in the category any type of complementary medicine. 

Thus, the variable any type of complementary medicine is not the sum of all types of CM but gives the information how many participants used at least one CM therapy in the past 12 months. We have clarified this information in the legend of the table 1 (line 215, p.12)

4. In table 3: Supplemental health insurance for complementary medicine: Yes (10815), No (5600) and Don’t know(2292) should total to N= 18,832 but it is not. Please clarify why

There were 125 missing data (i.e. 125 participants did not answer the question). We clarified this information in the legend of the table 3 (line 225, p.13).

5. Another important aspect is the duration of use of CM, some individuals use it for short duration for example for some short duration issue like 1- 2 days(constipation, diarrhoea, pain ) for 2-3 days. Was there any criteria for inclusion of data based on duration of use of CM.

Unfortunately, the Swiss Health survey did not provide any information on frequency and duration of CM use. We acknowledged this lack of information as a limitation of the study (lines 452-453).

6. The use of CM it is most of the time not taken after consultation from GP or other expert but is usually taken through recommendation by family members, friends or other acquaintances. Highlighting these sources other than experts would have been interesting as it may have safety issues

We agree with the reviewer on this aspect. Unfortunately, such detailed information was not available in the Swiss health survey and thus we could not further elaborate on it.

Reviewer #2 

1. Thank you for the opportunity to review this work. The 2017 and 2012 survey data were large but may be outdated. The authors should be applauded for the effort to conduct add-on questionnaire with a relatively good response rate. Nonetheless, the ‘85.1%’ was based on the 18,832 as the numerator and the 22,134 (but not 43,769) as the denominator. Please clarify and add an overview of the Swiss Health Survey in relationship with this particular study.

The 2017 Swiss health survey is the most recent national health survey currently available in Switzerland. The most recent Swiss health survey dates from 2022 and is currently being conducted. This point has been clarified in the manuscript accordingly (lines 102-103, p. 6). All the details concerning the Swiss Health Survey in relationship with our study are described in lines 93-108, p. 5-6. 

2. The coverage by mandatory basic health insurance seems to be intuitively the main factor of the increased CM use. As such, the analysis and/or discussion should differentiate the marginal effect of individual preferences, given the insurance coverage. When was the CM covered (before 2012 / between 2012-2017 / after 2017)? Table 3 presented the ‘Supplemental health insurance for complementary medicine’ as a separate finding. Actually, this point could be relatively easy to address by analyzing secondary annual claim data that should be available in the developed high-income country context like Switzerland. This suggested approach not only could provide the ‘revealed preferences’ of Swiss individuals, but also reduce the unreliable operational definitions of the CM and conventional therapies perceived by the respondents as pointed out below.

CM coverage by basic health insurance was suppressed in 2005, and started again in 2012. Therefore, in 2017 and in the comparison data of 2012, there were no change considering the reimbursement. Meanwhile, only some CM delivered by certified physicians are reimbursed by the mandatory basic health insurance. Meanwhile, it was not possible to differentiate among respondents if they used a CM reimbursed by mandatory basic health insurance or not. For example, a respondent who used TCM could have used it with a physician whose service was reimbursed by the mandatory health insurance or a therapist reimbursed by a supplemental health insurance or by none. Therefore, a secondary analysis did not allow for better information on this specific point. In the Discussion section, we discussed the role of regulations in terms of inclusion of CM in health insurance in European countries with the highest rates of prevalence (lines 406-410, p.27). Additionally, we have clarified the fact that in Switzerland CM can be covered by mandatory basic health insurance or private supplemental health insurance (lines 418-422, p.27). 

3. Line 110-114 presented a series of therapies without clear definitions to the readers (and to the respondents). Given the fact that these therapies have been covered by the mandatory basic health insurance, they should be clearly defined, along with whether each of the therapies is fully reimbursable and whether out-of-pocket or copayment is required.

In the written questionnaire, respondents were asked whether they had used the following therapies in the past 12 months: osteopathy (yes/no), naturopathy (yes/no), homeopathy (yes/no), herbal medicine (yes/no), acupuncture (yes/no), shiatsu or reflexology (yes/no), TCM (yes/no), Ayurveda (yes/no), or other therapies such as kinesiology, Feldenkrais, autogenic training, neural therapy, bioresonance therapy and anthroposophic medicine (yes/no). No additional information was available for the participant. Most of these therapies are not covered by the mandatory basic health insurance. Covered are currently: anthroposophic medicine, homeopathy, herbal medicine, and traditional Chinese medicine (TCM)) if delivered by a certified physician. Private complementary insurances cover these therapies if delivered by an accredited therapist. Level of coverage of CM therapies varies between private insurances. As previously explained, it was not possible to differentiate in the questionnaire whether the user was reimbursed by a mandatory or a private health insurance, or out of pocket. This information has now been added and clarified in the section Introduction (lines 58-61, p.4) and in the Discussion section (lines 418-422, p.27).

4. Line 114-118: What are the additional benefits of the arbitrary and non-standard CM categorization?

There is no official CM categorization nowadays. The four CM categories of this study were chosen according to the coverage by the mandatory basic health insurance. As mentioned in lines 416-422 (p. 27), in Switzerland, the mandatory basic health insurance covers anthroposophic medicine, homeopathy, herbal medicine and TCM if delivered by certified physicians. As the number of anthroposophic medicine users was low, we decided to merge it with ‘other CM therapies’ category. 

5. Line 120-123: Conventional health care is poorly defined and, therefore, could not be referred to as a control group for the research question of this study.

We adopted the categories as defined by the Swiss Federal Statistical Office in the Swiss Health Survey. Conventional health care includes all general practitioners and other medical specialists graduated from the university of medicine in Switzerland or who had obtained a recognition of a foreign diploma. 

The manuscript has been modified accordingly (lines 125-126, p. 7). 

6. Line 130: Was the age variable collected as a categorical variable as presented here? If not so, how do the authors justify these arbitrary categories? Please clarify this point with the other ‘ever-continuous-but-now-categorical’ variables as well.

We adopted the categories as defined by the Swiss Federal Statistical Office in the Swiss Health Survey.

7. Line 163: Is ‘last cannabis consumption’ part of the Lifestyle indicators of the Swiss Health Survey? If not, please differentiate which variables are from those that are not.

Yes, ‘last cannabis consumption’ was defined as a lifestyle indicators as part of the the Swiss health survey.

---

## [Decision Letter · Decision Letter 1]

26 Aug 2022

Prevalence, specific and non-specific determinants of complementary medicine use in Switzerland: data from the 2017 Swiss Health Survey

PONE-D-22-07999R1

Dear Dr. Meier-Girard,

We’re pleased to inform you that your manuscript has been judged scientifically suitable for publication and will be formally accepted for publication once it meets all outstanding technical requirements.

Kind regards,

Sergio A. Useche, Ph.D.

Academic Editor

PLOS ONE

Additional Editor Comments (optional):

The authors have done a good job in responding to the reviewers’ remaining comments. The paper can be now accepted for publication. la 

Reviewers' comments:

Reviewer's Responses to Questions

**Comments to the Author**

1. If the authors have adequately addressed your comments raised in a previous round of review and you feel that this manuscript is now acceptable for publication, you may indicate that here to bypass the “Comments to the Author” section, enter your conflict of interest statement in the “Confidential to Editor” section, and submit your "Accept" recommendation.

Reviewer #2: All comments have been addressed

2. Is the manuscript technically sound, and do the data support the conclusions?

Reviewer #2: Yes

3. Has the statistical analysis been performed appropriately and rigorously? 

Reviewer #2: Yes

4. Have the authors made all data underlying the findings in their manuscript fully available?

Reviewer #2: Yes

5. Is the manuscript presented in an intelligible fashion and written in standard English?

Reviewer #2: Yes

6. Review Comments to the Author

Reviewer #2: All of my comments have been satisfactorily addressed. Nonetheless, the manuscript still requires formatting edits.

7. PLOS authors have the option to publish the peer review history of their article (what does this mean?). If published, this will include your full peer review and any attached files.

Reviewer #2: **Yes: **Assoc. Prof. Dr. Krit Pongpirul, MD, MPH, PhD.

---

## [Editor Report · Acceptance letter]

5 Sep 2022

PONE-D-22-07999R1 

Prevalence, specific and non-specific determinants of complementary medicine use in Switzerland: data from the 2017 Swiss Health Survey. 

Dear Dr. Meier-Girard:

I'm pleased to inform you that your manuscript has been deemed suitable for publication in PLOS ONE. Congratulations! Your manuscript is now with our production department. 

Kind regards, 

on behalf of

Dr. Sergio A. Useche 

Academic Editor

PLOS ONE